# Importance of Mid-Infrared Spectra Regions for the Prediction of Mastitis and Ketosis in Dairy Cows

**DOI:** 10.3390/ani13071193

**Published:** 2023-03-29

**Authors:** Stefan Gruber, Lisa Rienesl, Astrid Köck, Christa Egger-Danner, Johann Sölkner

**Affiliations:** 1Institute of Livestock Sciences, University of Natural Resources and Life Sciences, Vienna (BOKU), Gregor-Mendel-Straße 33, 1180 Vienna, Austria; 2ZuchtData EDV-Dienstleistungen GmbH, Dresdner Straße 89/19, 1200 Vienna, Austria

**Keywords:** dairy cow, mastitis, ketosis, machine learning, variable selection

## Abstract

**Simple Summary:**

Mid-infrared spectral regions are routinely used to predict milk components, such as fat and protein, but can also be assisting to detect diseases in dairy cows, including mastitis and ketosis. However, not all wavelengths carry relevant information and are therefore not necessary for prediction. Thus, the objective of this study was to identify which wavelengths are of particular importance for prediction of mastitis and ketosis. Results indicate that important wavenumbers varied across different traits. For the prediction of mastitis, 23 wavelengths were selected as highly relevant, while for the prediction of ketosis, 61 wavelengths were particularly important. Thus, this implies that it is reasonable to select the most important spectral regions for the prediction of mastitis and ketosis in dairy cows.

**Abstract:**

Mid-infrared (MIR) spectroscopy is routinely applied to determine major milk components, such as fat and protein. Moreover, it is used to predict fine milk composition and various traits pertinent to animal health. MIR spectra indicate an absorbance value of infrared light at 1060 specific wavenumbers from 926 to 5010 cm^−1^. According to research, certain parts of the spectrum do not contain sufficient information on traits of dairy cows. Hence, the objective of the present study was to identify specific regions of the MIR spectra of particular importance for the prediction of mastitis and ketosis, performing variable selection analysis. Partial least squares discriminant analysis (PLS-DA) along with three other statistical methods, support vector machine (SVM), least absolute shrinkage and selection operator (LASSO), and random forest (RF), were compared. Data originated from the Austrian milk recording and associated health monitoring system (GMON). Test-day data and corresponding MIR spectra were linked to respective clinical mastitis and ketosis diagnoses. Certain wavenumbers were identified as particularly relevant for the prediction models of clinical mastitis (23) and ketosis (61). Wavenumbers varied across four distinct statistical methods as well as concerning different traits. The results indicate that variable selection analysis could potentially be beneficial in the process of modeling.

## 1. Introduction

MIR spectroscopy is one of the most used methods in food analysis regarding its authenticity and quality determination and finds application in agricultural livestock farming, for instance dairy production with cows, sheep, or goats [1,2,3]. While MIR spectra in dairy cows are routinely used for the prediction of major milk components, such as fat, protein, lactose, or urea, various components of milk with lower concentration can also be predicted from MIR spectral data [4], such as several minerals, fatty acids, or lactoferrin contents of milk [5]. Other studies focused on the prediction of methane emission [6], daily energy intake, feed intake or efficiency [7,8], milk differentiation regarding feeding systems [9,10], or pregnancy status [11,12]. Recently, quite some research has been conducted addressing predictive models based on MIR spectra to detect economically relevant diseases, including prognosis of clinical mastitis [13,14], ketosis [15], ketone bodies [16], or lameness [17].

The current study focuses on the prediction of clinical mastitis and clinical ketosis, as these two pathologies have high prevalence and relevance in the dairy sector [18]. Mastitis is characterized as an infection of the bovine mammary gland, which is induced by pathogens, for example *Staphylococcus aureus* [19,20]. It is the most common disease in dairy cows and entails several short- and long-term consequences for the diseased cow, including a decrease in milk quantity by 375 kg on average per clinical incident or five percent throughout a total lactation period [21], increased culling rates by 1.5 to 5 times [21], and an inferior qualitative constitution of milk and poorer fertility, as cows suffering from mastitis are open 30 days longer on average than healthy cows [19]. Overall, mastitis events lead to adverse repercussions on the welfare situation of the animal [20] and poorer economic output [22], estimated with deficits per cow ranging between 17 EUR and 198 EUR [23], also depending on the specific causative pathogen [19]. Mastitis can be diagnosed by veterinarians, through direct mastitis tests in laboratories, such as microbiological detection, or indirect mastitis tests, such as the Modified California Mastitis Test, which is a cow-side indicator for heightened somatic cells in milk [24,25]. For mastitis monitoring, indicators such as somatic cell count (SCC) can also be used. In addition, the application of MIR spectral data could be an auxiliary method for mastitis risk detection, as several studies already outlined the potential [13,26,27]. Ketosis is another prevalent and central metabolic disease mostly occurring during the first weeks of lactation in dairy cows. Cows suffering from ketosis are predisposed for other metabolic diseases, including metritis, mastitis, displaced abomasum, or lameness [28]. Moreover, daily milk yield decreases, the reproductive potential is restricted, and the risk of culling increases, which has severe economic consequences for farm profitability and for animal welfare [29]. For ketosis screening, MIR spectral analysis could be a tool provided to farmers in addition to veterinarian diagnoses. Some studies already tried to assess subclinical ketosis with prediction models based on milk components, such as ketone bodies and fatty acids derived from MIR spectra [16,30,31]. In these studies, prediction accuracies for ketone bodies were consistently high, and thus, valuable for ketosis screening. Werner et al. [32] predicted clinical ketosis based on MIR spectra directly and achieved good specificities (0.84) and moderate sensitivities (0.72).

The development of the prediction models can either be based on the entire MIR spectrum or merely on specific regions of the spectra which were identified to be relevant and the absorbance of which is generally reproducible [33]. While some studies recommend excluding certain wavelength ranges in order to obtain enhanced accuracy and robustness for the model [33,34,35], others indicate that regions that were suggested to be excluded may still provide information [36,37] and do not recommend to exclude them.

The objective of this study was to identify which individual regions of the MIR spectra are of importance for the prediction of bovine clinical mastitis and clinical ketosis. Furthermore, it was aimed to compare four methods of prediction, partial least squares discriminant analysis (PLS-DA), support vector machine (SVM), least absolute shrinkage and selection operator (LASSO), and random forest (RF). Their performance in the prediction of mastitis and ketosis was analyzed and comparative variable importance in projection (VIP) analysis performed. For reference, VIP analysis was also performed for standard milk components (fat, protein, lactose, and urea) routinely measured in the Austrian milk recording system.

## 2. Materials and Methods

### 2.1. Data and Data Preparation

This study was conducted based on two main datasets: dataset I, which was applied for the prediction models of mastitis and various milk components, and dataset II, which was utilized for the ketosis prediction model. Both datasets were provided by Zuchtdata GmbH and originated from the Austrian milk recording system and associated health monitoring system (GMON) [38]. All data were collected in the period of July 2014 to January 2020 for Fleckvieh (dual purpose Simmental), Holstein, and Brown Swiss cows. Both datasets were composed of test-day milk data and respective diagnosis data. The Austrian milk recording system has between 9 to 11 routine test-day recordings annually, corresponding to a test-day interval of 33–41 days [39]. Test-day milk data contained the following information: date of test day, animal ID (encrypted), herd ID (encrypted), region, breed, parity, calving date, days in milk, milk yield, fat%, protein%, lactose%, urea, and SCC from cows between 3 and 365 days in milk. MIR spectral data which were analyzed utilizing spectrometers (FOSS, Hillerod, Denmark) in official laboratories were included for the corresponding test days. Each of the 1060 MIR spectral variables indicates an absorbance value of infrared light at a specific wavenumber (926 cm^−1^ to 5010 cm^−1^) and was standardized to generate a uniform basis across various spectrometers and time periods [40]. Diagnosis data contained diagnoses of clinical (acute or chronic) mastitis (dataset I) or clinical ketosis (dataset II), and the corresponding dates of diagnosis. Only data from validated farms, where at least 75% of diagnostic data were electronically submitted by veterinarians, were comprised. Table 1 gives an overview of the number of records in datasets I and II.

For data cleaning, merging and some basic data preparation of both datasets SAS software was applied [41]. In a first step, test-day data were merged with ‘adjoining’ mastitis or ketosis diagnoses from the GMON system. Therefore, test-day records were linked to the respective mastitis or ketosis diagnoses which were detected in the period of 21 days prior and 21 days after the test day. The specific time window of ±21 days was chosen, so that almost any diagnosis could be assigned to a test-day [14]. If a cow had no recorded mastitis or ketosis diagnosis in this particular time window, the test-day result was identified as healthy.

Subsequent data preparation was then carried out in RStudio [42]. First, derivatives of all MIR spectral variables were taken, according to the following equation, as recommended by several pertinent studies [4,6,43,44]:dx(*n*) = x (*n*) − x (*n* + 4). 

The resulting 1056 MIR variables were appended to the associated test-day records. The two final datasets consisted of 742,926 records (dataset I) and 341,698 records (dataset II), respectively, each having information on test-day data, associated MIR variables, and corresponding mastitis (9958) or ketosis (1391) diagnoses.

### 2.2. Prediction Models

The prediction models for both traits, mastitis, and ketosis, were designed equally regarding model settings. Due to the purpose of the study, the models were always based on 1056 MIR spectral variables and diagnosis data (mastitis or ketosis), which were recorded in a time frame of at most ±21 days from test day. The entire dataset was randomly divided by farm without replacement into two subsets, calibration and validation, utilizing the function ‘sample’ (base R package) [42]. In this way, the cows from the calibration subset did not originate from the same farm as the cows in the validation subset. The calibration set contained 70% of the data and the validation set contained 30% of the data. Furthermore, random down sampling was applied to equalize healthy and diseased (mastitis or ketosis) cases in the final calibration set, as the incidence of clinical mastitis and ketosis diagnoses is quite low (mastitis—1.3%, ketosis—0.4%). On basis of this data preprocessing, final prediction models for mastitis and ketosis were developed based on four different statistical methods as described hereafter. For certain parameters of individual methods, default settings were applied, if not indicated differently. Moreover, detailed characteristics of all applied methods are elaborated in Hastie et al. [45].

Partial least squares discriminant analysis (PLS-DA) works with a discrimination procedure relying on new variables built based on partial least squares. New variables are linear associations of original predicting variables in combination with the dependent class variable [45]. The ‘caret’ package in R [46] was applied with following parameters: model tuning with the function ‘trainControl’ and a 10-fold cross validation, number of latent variables set automatic with a maximum of 70 to prevent overfitting, differentiation based on class probability (threshold 0.5) and centered and scaled MIR spectral data [13].

Least absolute shrinkage and selection operator (LASSO) was applied. This approach pertains to the group of logistic regression methods, which shrink regression coefficients of certain variables to zero, building their class predictions based on a subset of variables [45]. For the prediction model, the function ‘glmnet’ from the R package ‘glmnet’ [47] was utilized with the parameters alpha 1 and the family set to binomial for class prediction. Penalty parameter lambda was user specified via ‘assess.glmnet’ with 0.008, as this achieved good model accuracy and limited the number of selected variables to a reasonable number.

The machine learning algorithm support vector machine (SVM) was considered. SVM can enhance the space for features to infinite by using kernels and aims to divide classes in this space through seeking for linear boundaries. This allows the classification to be more flexible, as in the actual wavelength space non-linear frontiers can be utilized [45]. For implementation the R package ‘e1071′ [48] was used. The function ’svm’ was applied with a radial kernel and default parameter settings [49].

The ensemble method random forest (RF) was chosen. For classification, RF produces numerous decision trees which are able to embrace complex interactions by creating branches based on root nodes which follow certain partition rules depending on input variables. Each tree provides one class label, which are averaged across the entire forest to receive a final class decision [45]. The R package ‘randomForest’ [50] was applied using its function ‘randomForest’. The number of trees was maximized to 1000 and the number of wavelengths randomly sampled at each split was set to 32 (number of wavelenghts^0,5^). Parameter tuning was done by ‘tuneRF’ based on the implemented estimation of Out-of-Bag error [51].

For indication of accuracies in the mastitis and ketosis prediction models, the following indicators were used: sensitivity, proportion of correct predicted disease cases; specificity, proportion of correct predicted healthy cases; balanced accuracy, mean of sensitivity and specificity; and AUC, area under the receiver operating characteristic (ROC) curve. The AUC value is a global indicator for discrimination accuracy and principally specifies the area under the ROC curve. The ROC graph portrays the sensitivity on the *y*-axis and 1-specificity on the *x*-axis. The AUC value can take values between 0 and 1 with following interpretation: <0.5—not useful; 0.5—no discrimination; and 1—perfect discrimination [52].

Prediction models for the milk components fat%, protein%, lactose%, and urea mg/L, were all developed on the basis of 1056 first-derivative MIR spectral variables in combination with the respective milk component to be predicted. The values for the four milk components were determined in official laboratories in the framework of the Austrian milk recording system utilizing a spectrometer (FOSS, Hillerod, Denmark). First, 200,000 records were randomly selected from dataset I by application of the function ‘sample’ of the base R package. Subsequently, the dataset was split into half, a calibration and a validation subset by farm with no replacement with the function ‘sample’ [42]. The prediction models were then established using the method partial least squares (PLS) regression. This regression approach is based on constructing new variables via linear associations of original variables and the dependent variable. New variables are built based on partial least squares and finally used for regression [45]. The ‘caret’ package in R [46] was applied with the settings described above.

Model fits of milk component predictions were evaluated by two main indicators: root mean square error (RMSE), standard deviation of residuals; and R-squared (R^2^), coefficient of determination, or proportion of the variance of the milk component to be predicted, which can be explained by the spectral variables [53].

In Figure 1, a graphical illustration provides an overview on the general structure of the prediction models for mastitis and ketosis. Analyses with all prediction models were replicated 10 times. For the evaluation of different statistical methods, model fit indicators were compared in pairs for significance applying *t*-tests with a Bonferroni–Holm correction (*p* < 0.05).

### 2.3. Variable Importance in Projection (VIP) Analysis

VIP analysis was applied in the framework of each statistical method, to identify the most important wavelengths for the prediction of the respective trait. The main objective of a VIP analysis, also called feature selection analysis, is to indicate the relevance of independent variables for the prediction of a dependent variable [34]. Depending on the statistical method, distinct approaches for the evaluation of VIP were applied, as elaborated below.

To assess variable importance in the PLS-DA and PLS regression model, the filter ‘varImp’ was applied, which outputs a complete VIP ranking. The input variables are ranked according to sums of absolute regression coefficients weighed based on the decrease of the sums of squares across the amount of components in the PLS model [54,55].

LASSO is the only method considered in this study, which performs automatic feature selection, through shrinking coefficients for certain variables to 0—either because they do not carry relevant information or they are highly correlated to informative variables, already assigned with a non-zero coefficient [56]. For the evaluation of the most important variables, the model was repeated 100 times and the frequency of each variable was selected via the LASSO algorithm throughout these 100 runs, calculated as the VIP score [57].

In general, SVM does not directly perform feature selection, but assigns certain weights to every input variable for classification. These weights quantify the relative importance of the individual variable and can, therefore, be taken into account for variable importance [58,59]. Thus, wavelengths were ranked according to their weight assigned by SVM.

For VIP analysis in RF the extractor function ‘importance’ was used, which provides a direct variable importance ranking. This importance of each variable is determined based on the average accuracy decrease if the specific variable is precluded as a predictive variable [51,54].

With the result of a VIP score, each individual wavelength indicates its relevance for the prediction of the dependent target variable (mastitis, ketosis, or milk component) in the respective statistical method. Based on this variable importance score, wavelengths were sorted in descending order of importance, obtaining variable importance rankings for every target variable from every statistical method and corresponding feature selection algorithm. Proceeding from this, comparisons of respectively 1% (11 MIR spectral variables) and 10% (106 MIR spectral variables) of the most important wavelengths for the prediction of mastitis and ketosis with four different statistical methods were conducted. The most important 1% and 10% of wavelengths for the prediction of mastitis, ketosis, fat, protein, lactose, and urea based on PLS-DA and PLS regression were contrasted as well. For a wavelength to be considered of particular relevance, it has to be selected by various approaches for VIP identification concomitantly [10,34]. Thus, wavelengths which were commonly included in the most important 10% of at least three distinct methods were considered particularly important for the respective trait (mastitis or ketosis).

## 3. Results

### 3.1. Performance of Prediction Models for Various Milk Components

In general, prediction models for various milk components achieved highly accurate model results (Table 2). The models for determination of protein% obtained a low RMSE (0.0465) and a high R^2^ (0.9859), similar to those for lactose% with a RMSE of 0.0229 and a R^2^ of 0.9824. Both R^2^ being nearly 1 in validation indicates a very precise prediction of the respective component. This is similar for fat% prediction, with validation RMSE of 0.1651 and R^2^ of 0.9484, although it is less accurate than prediction models for protein% and lactose%. Determining urea content was the least accurate in comparison, still showing reliable performance. The results were very similar for the calibration dataset.

### 3.2. Model Performances for Mastitis Prediction

Table 3 presents an overview of various indicators regarding model performance in mastitis prediction based on the four statistical methods considered. When comparing calibration and validation, all four methods performed better in model calibration and showed more significant differences between methods. SVM was significantly the best in calibration for all performance indicators. Specificities were substantially higher than sensitivities for all methods in validation and calibration. On the validation dataset, SVM and LASSO achieved significantly greatest levels for balanced accuracies (both 0.602) and AUC values (0.641, 0.640). As for sensitivities, SVM, PLS-DA, and LASSO did not differ significantly, while RF performed significantly the worst (0.525). Specificity was significantly better when applying RF (0.653) and SVM (0.648) compared to the two remaining methods. PLS-DA had a significantly lowest specificity (0.597), AUC (0.612), and balanced accuracy (0.581). Overall, prediction models for mastitis prediction attained moderate accuracy.

### 3.3. Model Performances for Ketosis Prediction

In Table 4, model performance indicators regarding ketosis prediction are outlined for the four statistical methods. In general, applying SVM generated significantly best model performance (sensitivity, specificity, and balanced accuracy) on the calibration dataset. All methods performed better in calibration as opposed to validation. In validation, LASSO obtained significantly highest level for specificity (0.823), while between the other three methods, no significant difference was observed. LASSO was also significantly better than SVM and RF concerning AUC value (0.877) and balanced accuracy (0.807), and on the same level as PLS-DA (0.870, 0.803). Regarding sensitivities, no significant differences were observed across PLS-DA (0.796), LASSO (0.791), and SVM (0.784). RF had significantly lowest sensitivity (0.755) and balanced accuracy (0.779). Generally, ketosis prediction models achieved good prediction accuracies.

### 3.4. Comparison of MIR Spectral Variables across Various Traits

In Figure 2, 1% and 10% of most important variables selected by PLS-DA (mastitis and ketosis) and PLS regression (fat%, protein%, lactose%, and urea) are depicted. Comparing them, it can be inferred that selected MIR spectral variables varied depending on the specific trait. Importantly, ten percent of wavelengths for milk components are exclusively prevalent in the spectral range from 948.81 cm^−1^ to 2992.98 cm^−1^, wherein selected wavelengths for protein% and lactose% were not located in between 1700.91 cm^−1^ and 2580.29 cm^−1^. The same applies for the most relevant 10% of wavelengths for ketosis prediction. Important spectral regions for mastitis can also be detected in upper areas of the MIR spectrum (>3016.12 cm^−1^) compared to the remaining traits. One percent of the most important wavelengths for each trait are located within the most relevant region recommended by Grelet et al. [33]. The predominant number of most relevant 10% of spectral regions for milk components and ketosis was discovered in this region as well, while the most important 10% of regions in mastitis prediction are located both within these regions but also notably outside.

### 3.5. Importance of MIR Spectral Variables for Mastitis Prediction

Figure 3 illustrates a selection of 11 (1%) and 106 (10%) most important wavelengths in each of the four methods applied for the prediction of mastitis. Analyzing this post hoc, it was noticed that out of the total 302 different wavelengths, certain regions of the MIR spectrum were selected by several methods simultaneously, indicating that they are more likely influenced when a cow is diagnosed with clinical mastitis. In total, 23 wavelengths, allocated in the region from 964.23 to 4269.63 cm^−1^, were included in the most important 10% of at least three methods and were, thus, considered as particularly relevant. Only five were selected by all four methods.

In Table 5, pairwise conformance of 1% and 10% of selected MIR spectral regions between two specific methods are outlined. Conformance in selected 106 variables ranged for all pairwise comparisons between 23.6% (PLS-DA–RF) and 27.4% (SVM–RF), except PLS-DA versus SVM, where the level of conformance was 18.9%. Considering respective 11 spectral variables with the greatest importance in each method, they did coincide to a minor extent. Only PLS-DA and LASSO obtained an overlap of 4 (36.4%) wavelengths. The selected 1% of regions for PLS-DA, LASSO, and RF were predominantly in the region between 960.38 and 1450.21 cm^−1^, and all of them within the informative region according to Grelet et al. [33]. In total, 23 wavelengths were selected as highly relevant for the prediction of clinical mastitis.

### 3.6. Importance of MIR Spectral Variables for Ketosis Prediction

Figure 4 shows the 11 (1%) and 106 (10%) most relevant wavelengths for ketosis prediction for the applied methods. Evaluating the 248 wavelengths encompassing the 106 most relevant ones of all four methods, specific MIR spectral regions which were chosen by more than one method were identified to be more relevant for classification whether a cow suffers from clinical ketosis or is healthy. In particular, 61 MIR spectral variables (929.52–3019.98 cm^−1^) were included by at least three distinct methods, and hence, a specific relevance was attributed to them. Fifteen wavelengths were in the 10% selection of all methods.

As shown in Table 6, ten percent of the selected MIR spectral regions coincided quite considerably between subsequent pairs of methods: PLS-DA and SVM (50.0%), PLS-DA and RF (73.6%), and RF and SVM (59.4%). Pairwise comparisons of methods including LASSO obtained lower conformity across selected variables (19.8–24.5%). Almost the entire selection of 1% of the most relevant variables is located in the range of 933.38 to 1535.06 cm^−1^ and in the relevant information carrying region recommended by Grelet et al. [33]. Hence, conformance between three pairs of methods (PLS-DA–SVM, PLS-DA–RF, and SVM–RF) is high (>72.7%), only pairwise comparisons with LASSO were at a lower level (27.3%). Summarizing, 61 spectral regions were identified to be of particular importance for clinical ketosis prediction.

## 4. Discussion

### 4.1. Model Performances

This study investigated the suitability of MIR spectral data for the prediction of bovine clinical mastitis and clinical ketosis with different methods, as well as various milk components, and outlined particularly relevant wavelengths for predictions, respectively.

As was to be expected, the prediction models for various milk components obtained substantially higher accuracies (Table 2) than the developed models for mastitis and ketosis prediction. This is due to the fact, that prediction models were developed based on values for milk components, which were already determined based on spectral data in the framework of the Austrian milk recording system. Moreover, fat%, protein%, and lactose% exhibit typically high predictability based on MIR spectral data, as substantiated by relevant studies [60,61,62]. The prediction of urea content in milk was not as precise as the remaining three milk components, which is congruent with Melfsen et al. [62], who reported R^2^ values of 0.998 for fat%, 0.94 for protein%, 0.73 for lactose%, and 0.31 for urea in mg/L.

The models for prediction of clinical mastitis were least precise in the present study (Table 3), with AUC values ranging from 0.612 to 0.641 across different statistical methods. According to the definition from Simundic [52], this complies to a sufficient diagnostic accuracy. A recent study of Rienesl et al. [27] outlined better results for all accuracy parameters for the same ±21-day time window and predictions based on MIR spectra; however, they utilized a preselection of days in milk corrected spectral variables and additional fixed effects (parity, breed). Sensitivities and specificities reported from Dale et al. [26] were considerably better than those presented in this study as well, which is attributable to a days in milk correction, 212 selected MIR variables, and, most importantly, by a definition of the mastitis event deviating substantially from that of the current study.

Evaluation of the four different methods applied for mastitis prediction regarding accuracies in this study indicates that LASSO and SVM are suited best for mastitis prediction with AUCs greater than 0.640. Particularly noticeable is that SVM was best on the calibration dataset, highlighting its capability of a more flexible classification procedure due to using radial kernels [45]. The LASSO algorithm providing an automatically implemented feature selection approach and achieving best model accuracies highlights the importance of variable selection for model development. All methods obtained considerable higher validation specificities (0.597–0.653) than sensitivities (0.525–0.567). This is reasonable, as mastitis cases in this study were defined based on veterinarian diagnosis solely, which appears to be a limitation in this context. Mastitis cases which were not treated by a veterinarian, and hence, not recorded in the GMON system, are surely present in the data. The developed prediction models potentially labeled these undetected mastitis events as mastitis cases, whereas they were identified as healthy in the dataset, leading to a sensitivity decrease in the prediction models. By defining mastitis and healthy cases based on clinical assessment, such restraints could be prevented to a great extent. In this study, different methods showed significant differences in model performance of mastitis prediction. However, comparing results with pertinent studies reveals that input variables have a greater impact than applied methods, which is consistent with Young et al. [63].

The prediction models reported for clinical ketosis (Table 4) in the actual work were noticeably better in model performance than those for mastitis, yet not as precise as milk component prediction models. Considering Simundic’s [52] definition, all four applied methods attained very good diagnostic accuracy, as AUC values were in the range from 0.856 to 0.877. As far as comparable literature is concerned, nearly no literature exists on the detection of clinical ketosis directly via MIR spectral data. Most studies determined subclinical ketosis risk based indirectly on MIR spectra through consideration of specific components (e.g., ketone bodies) being predicted based on MIR spectral data [16,31]. De Roos et al. [31] outlined higher specificities and lower sensitivities, while sensitivities and specificities reported by van Knegsel et al. [16] were both greater and lower depending on the definition of subclinical ketosis. To our knowledge, KetoMIR2 is the only reference also predicting clinical ketosis considering MIR spectra directly [32]. However, it has to be taken into consideration that KetoMIR2 was developed on 212 selected MIR spectral variables, and time windows were divergent to the ±21-day window in the current model (±14 days for ketosis, ±60 days for healthy) and included only cows during 5 to 120 days in milk. Sensitivity from KetoMIR2 was worse (0.72) than in the present ketosis prediction models, which is quite noteworthy. The stricter definition based on the ±14-day time window in that study would suggest possibly more precise predictions. Even though specificity (0.84) was better, balanced accuracy from KetoMIR2 was consistently lower than in all four ketosis prediction methods applied in the actual study. In previously quoted studies, observed sensitivities and specificities were not as balanced in contrast to present results. This may underline advantageous model development in this study, for instance finetuning of parameters or ketosis definition.

When comparing the four methods for ketosis prediction concerning model performance, it can be inferred that all four methods are able to predict clinical ketosis events at a proficient level. RF is the least promising, while the remaining three methods achieve nearly equal balanced accuracies in validation. LASSO obtained the best results for all accuracy indicators, in turn emphasizing the importance of feature selection. More considerable differences across all accuracy parameters were noticed in calibration, demonstrating varying capabilities of respective methods to adapt their calibration procedure to an individual dataset. Based on model accuracies, detection of clinical ketosis via MIR spectral data is very promising.

### 4.2. Selected MIR Spectral Variables

The evaluation of selected MIR spectral variables across various traits (Figure 2) highlighted that for each individual trait distinct wavelengths were of special importance. Therefore, MIR spectral regions carry individual information, potentially providing insights concerning certain traits corresponding to dairy cows.

The present study included all 1056 first-derivative MIR spectra as input variables in the prediction models and intentionally renounced variable preselection. This was done to potentially detect information carrying spectral regions also outside the commonly known informative regions. These regions are declared as water diluted, noisy, not ideally reproducible, or lacking relevant information, and therefore, often not included in developed models: 925 to 960 cm^−1^ [33], 1600 to 2800 cm^−1^ [33,35], and 3000 to 5000 cm^−1^ [1,33]. In fact, this study detected important regions for various traits in these areas, as visualized in Figure 2, Figure 3 and Figure 4, implying that information is also available in these parts of the spectrum. One reason why other studies did not report information in these areas of the spectra might be that VIP analysis was not applied on the entire MIR variable set [10,34,64]. They precluded certain regions due to water absorption or low reproducibility across spectrometers already in advance, so they did not even search for information in these areas. There is also not much relevant literature having tested variable importance regarding specific characteristics (e.g., mastitis or ketosis) explicitly, but rather determined informative regions from a chemical or technical point of view [33]. When evaluating different methods, it becomes obvious that PLS-DA was the only method not selecting a wavelength in the part of the spectra above 3922.5 cm^−1^. Especially LASSO and SVM assigned importance to several regions in this upper part of the spectra for ketosis and mastitis. This suggests that LASSO and SVM are able to find information in noisy regions, while PLS-DA cannot utilize these areas due to dilution, as also outlined by Andersen et al. [35]. Nevertheless, the results confirm that the predominant proportion of wavelengths carrying relevant information are located within the commonly recommended areas by Grelet et al. [33]. Hence, the results of the present study imply that studies aiming to predict certain traits for dairy cows, including mastitis and ketosis, should include spectral regions that go beyond the recommended ones by Grelet et al. [33], as relevant information is available there as well. In addition, variable selection analysis can be beneficial to identify a subset of selected spectral regions for final prediction models.

For the determination of all four milk components (fat%, protein%, lactose%, urea), the region containing most informative wavelengths was identified from 948.81 to 2992.98 cm^−1^ (Figure 2). This is basically consistent with relevant literature, suggesting uninformative regions above 3000 cm^−1^ [1,33].

VIP analyses for mastitis in this study demonstrated that important wavelengths are rather spread across the entire spectral range from 925.66 to 4909.88 cm^−1^ (Figure 3). This highlights the presence of information on mastitis outside commonly known spectral areas. However, conformances between 1% and also 10% most important variables of each method were quite low (Table 5), which could possibly indicate a limited availability of information on mastitis in the MIR spectra. Moreover, it implies that applying just one procedure for VIP analysis would not attain sufficient results, as outlined by Zhang et al. [34].

In Table 7, an overview of 23 particularly important wavelengths for mastitis prediction is provided. All these wavelengths were assigned with high relevance by at least three methods. Considering which methods tended to choose which wavelengths, it was noticeable that LASSO diverged most from the other methods. This algorithm tended to choose only one variable out of a cohesive spectral region, while the remaining three methods assigned importance to each of the wavelengths. For instance, four wavelengths from 1068.37 to 1079.94 cm^−1^ were all indicated as important by PLS-DA, SVM, and RF, while LASSO only selected 1072.23 cm^−1^. This limitation of LASSO has already been confirmed in the literature several times [57,65]. Taking this into account, the relevance of the 23 selected variables for mastitis prediction receives further substantiation.

Several studies already reported the connection between certain chemical bonds and corresponding MIR spectral wavelengths [37,68,69]. For instance, certain wavelengths identified as particularly important for mastitis prediction (Table 7) were already assigned to specific chemical bonds corresponding to lactose, carbohydrates, casein, phospholipids, triglycerides, lipids, water, and melatonin. For more detailed information concerning chemical bonds found at the respective wavelengths, we refer the reader to certain references listed in Table 7. Particularly interesting are the identified regions potentially being assigned to casein contents of milk (1068.37–1079.94 cm^−1^, 1234.22 cm^−1^), as a decrease in casein amount in milk is an indicator for mastitis [73]. A recent study conducted by Huang et al. [74] highlighted the usefulness of sodium and chloride contents in milk for detection of mastitis cases in dairy cows. The literature already outlined that molecules containing sodium chloride can be assigned to MIR spectra regions between 3150 and 3400 cm^−1^ and specifically outlined 3357 cm^−1^ and 3341 cm^−1^ [72]. This leads to the assumption that the wavelengths identified as particularly important in this region (3255.25 cm^−1^, 3340.1 cm^−1^, 3347.82 cm^−1^, 3359.39 cm^−1^) might represent the elevated sodium chloride content in milk from cows suffering from mastitis.

The evaluation of VIP analyses for ketosis in the present work showed that 10% of relevant wavelengths tended to be distributed in two principal areas: 925.66 to 2055.75 cm^−1^ and 2703.71 to 4967.73 cm^−1^ (Figure 4). Hence, it can be deduced that the region in between does not contain information on ketosis, which is reasonable, as this range of the spectra shows water absorption. Yet, it is also evident that information for ketosis is present in the noisy area. When contrasting selected regions concerning methods, it is interesting to see that PLS-DA, RF, and SVM tended to select more similar MIR spectral variables, while LASSO had substantially lower pairwise conformity of the selected wavelengths with the rest of the methods. Previously discussed reasons, such as avoidance of selection of neighboring spectra positions by LASSO, may hold here as well. Moreover, high conformance results (Table 6) indicate that important regions were not just selected randomly, and thus, information on clinical ketosis exists in the MIR spectrum.

Table 8 lists certain regions of the MIR spectra which were identified as particular important for the prediction of clinical ketosis (concomitantly selected by at least three methods). When comparing these outlined regions with the preselected regions Hansen [30] used for ketosis prediction, it shows that 56 out of the 61 selected wavelengths were located within the region from Hansen, providing an underpinning of these important variables. The fact that most of them are consecutive wavelengths is particularly considerable and strengthens their importance moreover. Here, again, it becomes obvious that LASSO did not choose every variable out of a correlated subgroup of spectra.

Other studies, as referenced in Table 8, already outlined which chemical bonds can be connected to which wavelengths. In this case, wavelengths reported as particularly relevant for ketosis prediction were attributed with chemical bonds originating from lactose, phospholipids, carbohydrates, proteins, amide, whey protein, melatonin, triglycerides, water, or casein. Concerning further information on detailed chemical bonds corresponding to detected important wavelengths for ketosis studies referenced in Table 8 can be insightful. One specific research found wavelength 1284.36 cm^−1^ to be of particular importance for the prediction of cows’ body weight [34]. Wavenumber 1284.36 cm^−1^ can be assigned to carbohydrates, which may suggest a potential connection in this context to a cows’ energy metabolism, as both ketosis and body weight are associated with that as well. Van Haelst et al. [75] discovered elevated concentrations of oleic acid (C:18 *cis*-9) in dairy cows milk indicate ketosis risk. Oleic acid can be assigned to certain spectra peaks, such as 2850 cm^−1^ [76]. As 2850.27 cm^−1^ was identified as particularly important for ketosis prediction in this study, this connection could be highlighted. It is also worth considering that wavelengths reported to contain information on ketone bodies [30,31,77] were not considered as particularly important in this study. This could be due to the fact that this work considered clinical ketosis diagnoses, so that cows suffering from subclinical ketosis were commonly among the cows labeled as healthy, which, thus, led to ketone bodies not assisting in discrimination.

## 5. Conclusions

This research evaluated the importance of MIR spectral variables for the prediction of bovine clinical mastitis and clinical ketosis. Four different statistical methods were assessed regarding model performance and determination of important variables. The results indicate that besides the most frequently applied PLS approach, other methods, particularly SVM and LASSO, owe considerable potential as prediction models for clinical ketosis and mastitis based on MIR spectral data. According to VIP analysis, it could be highlighted that different wavelengths were of particular importance for the prediction of mastitis, ketosis, and milk components, respectively, thus substantiating available information on specific traits of dairy cows in the milk MIR spectrum. A more detailed chemical evaluation of identified chemical bonds and comparison with possible indicators of mastitis and ketosis in milk could provide further insight into the identified particularly important regions of the MIR spectrum. The present study has contributed to enable a detailed insight into predictive models of clinical mastitis and clinical ketosis based on spectral data, thus assisting to better exploit the preventive potential of such models to timely detect these two pathologies.

## Figures and Tables

**Figure 1 animals-13-01193-f001:**
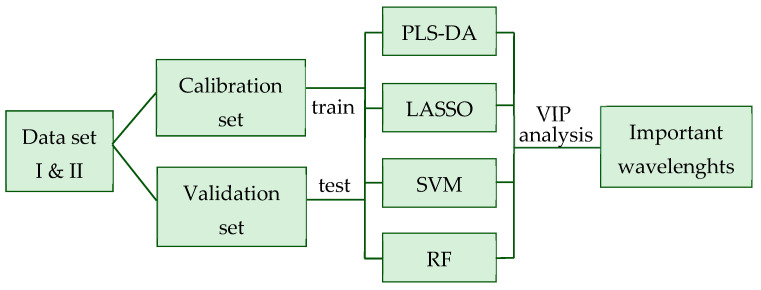
Overview on the general structure of the prediction models for mastitis and ketosis. PLS-DA = partial least squares discriminant analysis; LASSO = least absolute shrinkage and selection operator; SVM = support vector machine; RF = random forest; VIP = variable importance in projection.

**Figure 2 animals-13-01193-f002:**
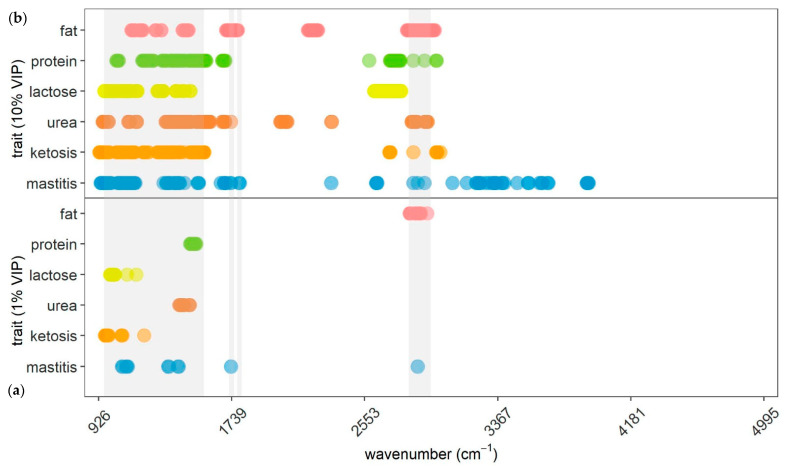
Representation of (**a**) 1% and (**b**) 10% of the most important wavelengths selected for the respective trait presented on the *y*-axis. Wavelengths for mastitis and ketosis were selected by partial least squares discriminant analysis and wavelengths for fat%, protein%, lactose%, and urea were selected by partial least squares regression. Gray areas indicating most informative areas according to Grelet et al. [33].

**Figure 3 animals-13-01193-f003:**
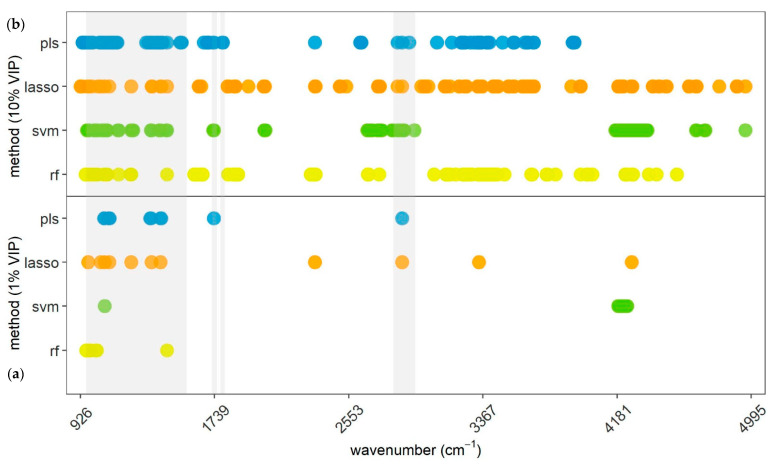
Representation of (**a**) 1% and (**b**) 10% of the most relevant wavelengths selected by the respective method shown on the *y*-axis for mastitis prediction. PLS = partial least squares discriminant analysis; LASSO = least absolute shrinkage and selection operator; SVM = support vector machine; RF = random forest. Gray areas indicating most informative areas according to Grelet et al. [33].

**Figure 4 animals-13-01193-f004:**
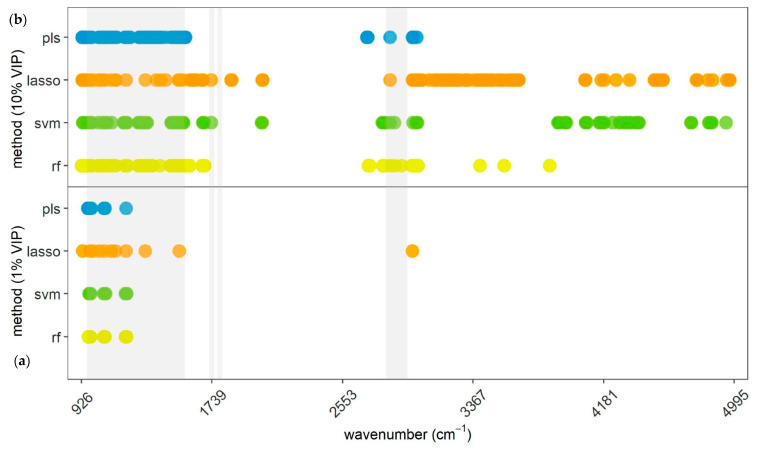
Representation of (**a**) 1% and (**b**) 10% of most important wavelengths for ketosis prediction selected by the respective method presented on the *y*-axis. PLS = partial least squares discriminant analysis; LASSO = least absolute shrinkage and selection operator; SVM = support vector machine; RF = random forest. Gray areas indicating most informative areas according to Grelet et al. [33].

**Table 1 animals-13-01193-t001:** Characteristics of dataset I utilized for the mastitis and milk component prediction models and dataset II applied for the ketosis prediction model.

Characteristic Dataset I	Records (*n*)	Characteristic Dataset II	Records (*n*)
Farms	2624	Farms	914
Cows	59,035	Cows	27,335
Fleckvieh	46,058	Fleckvieh	21,106
Brown Swiss	5332	Brown Swiss	2010
Holstein Friesian	7645	Holstein Friesian	4219
Test-day records	742,926	Test-day records	341,698
Mastitis records *	9958	Ketosis records *	1391

* test-day records in the time frame of 21 days prior/past mastitis/ketosis diagnosis.

**Table 2 animals-13-01193-t002:** Results of model performances for various milk components in calibration and validation datasets.

Milk Component	RMSEC	R^2^c	RMSEP	R^2^v
Fat%	0.1600 ± 0.0022	0.9516 ± 0.0012	0.1651 ± 0.0022	0.9484 ± 0.0013
Lactose%	0.0225 ± 0.0002	0.9828 ± 0.0003	0.0229 ± 0.0002	0.9824 ± 0.0003
Protein%	0.0465 ± 0.0009	0.9859 ± 0.0006	0.0465 ± 0.0009	0.9859 ± 0.0006
Urea mg/L	2.7880 ± 0.0177	0.8922 ± 0.0014	2.8165 ± 0.0162	0.8892 ± 0.0014

RMSEC = root mean square error of calibration; R^2^c = coefficient of determination in calibration; RMSEP = root mean square error of prediction; R^2^v = coefficient of determination in validation. Results represent mean values of 10 independent model runs ± standard deviation.

**Table 3 animals-13-01193-t003:** Results of model performances for distinct mastitis prediction methods during calibration and validation.

	Calibration			Validation			
Method	Sensitivity	Specificity	bal. acc.	Sensitivity	Specificity	bal. acc.	AUC
PLS-DA	0.613 ^a^ ± 0.008	0.656 ^a^ ± 0.008	0.635 ^a^ ± 0.007	0.566 ^a^ ± 0.018	0.597 ^a^ ± 0.015	0.581 ^a^ ± 0.008	0.612 ^a^ ± 0.008
LASSO	0.569 ^b^ ± 0.005	0.646 ^b^ ± 0.005	0.608 ^b^ ± 0.004	0.567 ^a^ ± 0.021	0.637 ^b^ ± 0.009	0.602 ^b^ ± 0.009	0.641 ^b^ ± 0.009
SVM	0.755 ^c^ ± 0.009	0.815 ^c^ ± 0.006	0.785 ^c^ ± 0.004	0.556 ^a^ ± 0.012	0.648 ^c^ ± 0.006	0.602 ^b^ ± 0.004	0.640 ^b^ ± 0.004
RF	0.616 ^a^ ± 0.003	0.600 ^d^ ± 0.003	0.608 ^b^ ± 0.002	0.525 ^b^ ± 0.018	0.653 ^c^ ± 0.012	0.589 ^c^ ± 0.005	0.624 ^c^ ± 0.004

PLS-DA = partial least squares discriminant analysis; LASSO = least absolute shrinkage and selection operator; SVM = support vector machine; RF = random forest; bal. acc. = balanced accuracy; AUC = area under the receiver operating characteristic curve. Results represent mean values of 10 independent model runs ± standard deviation. Distinct superscripts (^a^, ^b^, ^c^, ^d^) of results in the same column indicate significance of difference (Bonferroni–Holm method, *p* < 0.05).

**Table 4 animals-13-01193-t004:** Results of model performances for distinct ketosis prediction methods during calibration and validation.

	Calibration			Validation			
Method	Sensitivity	Specificity	bal. acc.	Sensitivity	Specificity	bal. acc.	AUC
PLS-DA	0.831 ^a^ ± 0.014	0.852 ^a^ ± 0.011	0.841 ^a^ ± 0.012	0.796 ^a^ ± 0.023	0.810 ^a^ ± 0.008	0.803 ^ab^ ± 0.011	0.870 ^ab^ ± 0.011
LASSO	0.814 ^b^ ± 0.007	0.836 ^b^ ± 0.007	0.825 ^b^ ± 0.004	0.791 ^a^ ± 0.027	0.823 ^b^ ± 0.007	0.807 ^a^ ± 0.012	0.877 ^a^ ± 0.009
SVM	0.894 ^c^ ± 0.007	0.907 ^c^ ± 0.009	0.901 ^c^ ± 0.005	0.784 ^a^ ± 0.027	0.803 ^a^ ± 0.011	0.793 ^b^ ± 0.010	0.863 ^bc^ ± 0.009
RF	0.813 ^b^ ± 0.007	0.834 ^b^ ± 0.011	0.823 ^b^ ± 0.007	0.755 ^b^ ± 0.020	0.804 ^a^ ± 0.014	0.779 ^c^ ± 0.007	0.856 ^c^ ± 0.010

PLS-DA = partial least squares discriminant analysis, LASSO = least absolute shrinkage and selection operator; SVM = support vector machine; RF = random forest; bal. acc. = balanced accuracy; AUC = area under the receiver operating characteristic curve. Results represent mean values of 10 independent model runs ± standard deviation. Distinct superscripts (^a^, ^b^, ^c^) of results in the same column indicate significance of difference (Bonferroni–Holm method, *p* < 0.05).

**Table 5 animals-13-01193-t005:** Conformance of selected MIR spectral variables for mastitis prediction according to pairwise comparisons of methods.

	PLS-DA	LASSO	SVM	RF
PLS-DA	-	25.5%	18.9%	23.6%
LASSO	36.4%	-	26.4%	24.5%
SVM	9.1%	9.1%	-	27.4%
RF	0%	9.1%	0%	-

PLS-DA = partial least squares discriminant analysis; LASSO = least absolute shrinkage and selection operator; SVM = support vector machine; RF = random forest. Values represent pairwise conformance of 1% (lower triangle) and 10% (upper triangle) of most important wavelengths in mastitis prediction between respective methods.

**Table 6 animals-13-01193-t006:** Conformance of selected MIR spectral variables for ketosis prediction between pairwise method comparisons.

	PLS-DA	LASSO	SVM	RF
PLS-DA	-	24.5%	50.0%	73.6%
LASSO	27.3%	-	19.8%	24.5%
SVM	72.7%	27.3%	-	59.4%
RF	81.8%	27.3%	90.9%	-

PLS-DA = partial least squares discriminant analysis; LASSO = least absolute shrinkage and selection operator; SVM = support vector machine; RF = random forest. Values represent pairwise conformance of 1% (lower triangle) and 10% (upper triangle) of the most important wavelengths in ketosis prediction between respective methods.

**Table 7 animals-13-01193-t007:** Overview of mid-infrared wavelengths considered as most important for mastitis prediction and respective references.

Wavenumbers (cm^−1^)	References *
964.23–979.66	[37,66,67,68,69]
995.09	[68,69]
1068.37–1079.94	[37,68,69]
1234.22	[68,69]
1357.64	[68,69]
1411.64	[68,69]
1450.21	[68,69,70]
2348.87	[69]
2738.42	[37]
2850.27	[37,68,69,71]
2877.27	[37,68,69]
3255.25	[37,68,69,72]
3340.1	[37,68,69,72]
3347.82	[37,68,69,72]
3359.39	[37,68,69,72]
4269.63	[37,68]

* Listed references provide further information regarding specific chemical bonds assigned to the respective wavelength regions.

**Table 8 animals-13-01193-t008:** Overview of mid-infrared wavelengths assigned as particularly important for ketosis prediction and respective references.

Wavenumbers (cm^−1^)	References *
929.52–937.23	[68,69]
956.52–991.23	[37,66,67,68,69]
1033.66–1041.37	[37,68,69]
1060.66–1079.94	[37,68,69]
1137.8	[37,68,69]
1195.65–1207.22	[37,68,69]
1284.36–1303.64	[37,68,69]
1322.93–1334.5	[68,69]
1484.92–1535.06	[37,68,69]
1550.49–1562.06	[68,69]
1681.62	[68]
2850.27	[37,68,69,71,75]
2989.12–2992.98	[37,68,69]
3016.12–3019.98	[37,69]

* Listed references provide further information regarding specific chemical bonds assigned to the respective wavelength regions.

## Data Availability

The data presented in this study are available on request from the corresponding author. The data are not publicly available due to privacy restrictions of the data provider and owner, the Austrian milk recording system (LKV Austria Gemeinnützige GmbH).

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
