# Peer review of "Importance of Mid-Infrared Spectra Regions for the Prediction of Mastitis and Ketosis in Dairy Cows"

_animals, 2023, doi:10.3390/ani13071193_

Round 1

Reviewer 1 Report

The present article shows the applicability of mid-infrared spectra to detect pathologies such as mastitis and ketosis. With the growing interest on precision livestock to adopt new technologies and methods in farm animals, the present article provides wide knowledge about MIR  and its use together with other statistical methods. I left some comments below hoping they can help the authors.   

Before the abstract, please, revise the author’s guidelines. The articles must include a simple summary.

Line 13: Specify what the authors mean by “…sufficient information.” Information about what?

Line 20: Instead of just saying that certain wavenumbers were identified, it would be helpful to mention which wavenumbers serve as predictors for mastitis and ketosis.

Line 24: Consider replacing “mid-infrared spectroscopy” with some of the other analysis or statistical methods that were used. I recommend this because MIR is already included in the title.

Lines 27-29: I consider that adding in which species MIR has been used is relevant. Or if is solely used in dairy cows.

Line 37: Add reference.

Lines 36-37: Since the aim of the study is explained below (lines 64-72), I recommend deleting this sentence. Also, it could be added why the authors decided to use mastitis and ketosis as study pathologies.

Lines 39-42: When mentioning “decrease in milk quantity”, “an inferior qualitative constitution of milk”, “increased culling rates” and “poorer fertility”, it would be good to include how much they decreased and some examples about the economic impact that mastitis have on dairy systems.

Lines 54-57: I suggest briefly including the results of the studies that already used MIR to predict ketosis (e.g., did they work? Were they accurate?)

Line 47: To highlight this information, the authors could mention that a correlation has been found between somatic cells counts and California test.

Line 72: If the authors had hypotheses for the present study, they can be added before Materials and Methods.

Line 103: Please, correct “therefore”.

Lines 240-242: Please, specify the RMSE and R2 values of the prediction models for protein% and lactose%.

Line 390: Consider including the results reported by Melfsen et al.

Lines 409-410: Would you consider the fact that mastitis cases were defined solely on veterinary diagnoses (understanding that you did not perform any clinical assessment?) a limiting factor of the present study? Or an element that could improve/decrease the specificity and sensitivity of the proposed methods? Please, try to discuss this.

Line 423: Include the number of a citation for De Roos et al. and van Knegsel et al.

Lines 476-479: I suggest adding a little discussion on the implication of the mentioned wavelengths. For example, according to the present results, further studies or all studies focusing on MIR-mastitis-ketosis should use these wavelengths?

Reviewer 2 Report

Overall, this manuscript contributes original research article. It is quite well-written, clear, relevant for the ruminant metabolism and prediction models for bovine clinical mastitis and clinical ketosis, which was presented in a well-structured manner. The objective of the study was clear. The study was appropriately designed, clearly explained research methodology and results, and correspondingly made a conclusion. The authors did a good job of the connection to previous/relevant studies and potential explanations in the discussion.

Author Response

Response to Reviewer 2 Comments

Point 1: Overall, this manuscript contributes original research article. It is quite well-written, clear, relevant for the ruminant metabolism and prediction models for bovine clinical mastitis and clinical ketosis, which was presented in a well-structured manner. The objective of the study was clear. The study was appropriately designed, clearly explained research methodology and results, and correspondingly made a conclusion. The authors did a good job of the connection to previous/relevant studies and potential explanations in the discussion.

Response 1: No suggestions for further improvement provided, therefore no changes made in the manuscript.

Reviewer 3 Report

The current manuscript is focused on mastitis and ketosis in dairy cattle, two diseases that have a very high prevalence regrettably.

The experimental design is sound, same can be said about the statistical analysis of the data. The datasets are rather impressive, with over 59,035 cows in the first experiment and over 27,000 cows for the second dataset. Moreover, the study is being focused on the three most important dairy/dual-purpose breeds found in Europe.

Just one observation: the manuscript lacks the simple summary;

And one suggestion: Given the complex approach, I strongly believe that a graphical abstract or a figure with the experimental design would help out readers a great deal.

I had not noticed any faults in the manuscript, and I believe that it meets the quality and relevance to be accepted for publication in the current form.

Author Response

Response to Reviewer 3 Comments

Point 1: Just one observation: the manuscript lacks the simple summary

Response 1: The simple summary has been by now added to the manuscript.

Point 2: And one suggestion: Given the complex approach, I strongly believe that a graphical abstract or a figure with the experimental design would help out readers a great deal.

Response 2: A graphical illustration to provide an overview on the general structure of the prediction models for mastitis and ketosis is included in the article now (Figure 1).

Reviewer 4 Report

Mastitis and ketosis are economically important diseases in the dairy industry. Majority of the efforts are focussed on the preventative aspects of these disease conditions. Predictive studies and risk factor analysis based on huge data sets provide a scientifically evidence based solutions to mitigate these diseases from dairy herds. Mid-infra-red spectroscopy is a valuable tool for analysis of milk samples from vulnerable herds and also as a routine random testing for disease surveillance. The topic has been relevant for the entire dairy industry worldwide. The study fulfils gaps in the field of early detection of clinical as well as subclinical phases of these two very important diseases. The large number of animals screened provide robustness and relevance to the results that will have wide implication for future studies and preventative strategies.

The manuscript has been nicely written, well organised in various subsections. Statistical analyses are excellent and well explained. Discussion is quite relevant and covers all aspects of the results obtained. I would suggest the authors to consolidate the results of each analysis succinctly to keep the reader rivetted to the reading. Outliers in the statistical analysis should have been discussed. The conclusions drawn should also point towards the preventative aspects of these diseases based on the usefulness of these predictive modelling of data. I hope the authors will include some of the suggestions. I am impressed with the presentation of the data analysis.

Author Response

Response to Reviewer 4 Comments

Point 1: I would suggest the authors to consolidate the results of each analysis succinctly to keep the reader rivetted to the reading.

Response 1: As suggested, a short consolidation of the results in each analysis was included in the end of the respective result chapters, where it was appropriate.

Point 2: Outliers in the statistical analysis should have been discussed.

Response 2: In general, there were no significant outliers in the statistical analysis part. Across the 10 repetitions of the prediction models the model fit indicators showed no outliers, which is shown by the low standard deviations across all models. Therefore there was no specific need for discussion.

Point 3: The conclusions drawn should also point towards the preventative aspects of these diseases based on the usefulness of these predictive modelling of data. I hope the authors will include some of the suggestions.

Response 3: A part highlighting the potential of MIR spectra-based prediction models for early detection of mastitis and ketosis has been added in the conclusion.